# An efficient Screening System in Yeast to Select a Hyperactive *piggyBac* Transposase for Mammalian Applications

**DOI:** 10.3390/ijms21093064

**Published:** 2020-04-26

**Authors:** Wen Wen, Shanshan Song, Yuchun Han, Haibin Chen, Xiangzhen Liu, Qijun Qian

**Affiliations:** 1Shanghai Cell Therapy Research Institute, Shanghai 201805, China; wenw@shcell.org (W.W.); songss@shcell.org (S.S.); hanyc@shcell.org (Y.H.); chenhb@shcell.org (H.C.); 2Shanghai Cell Therapy Group, Shanghai 201805, China

**Keywords:** non-viral transgenic vectors, *piggyBac* transposase, yeast, efficient screening system, transposition efficiency

## Abstract

As non-viral transgenic vectors, the *piggyBac* transposon system represents an attractive tool for gene delivery to achieve a long-term gene expression in immunotherapy applications due to its large cargo capacity, its lack of a trace of transposon and of genotoxic potential, and its highly engineered structure. However, further improvements in transpose activity are required for industrialization and clinical applications. Herein, we established a one-plasmid effective screening system and a two-step high-throughput screening process in yeast to isolate hyperactive mutants for mammalian cell applications. By applying this screening system, 15 hyperactive *piggyBac* transposases that exhibited higher transpose activity compared with optimized hyPBase in yeast and four mutants that showed higher transpose activity in mammalian cells were selected among 3000 hyPBase mutants. The most hyperactive transposase, bz-hyPBase, with four mutation sites showed an ability to yield high-efficiency editing in Chinese hamster ovarian carcinoma (CHO) cells and T cells, indicating that they could be expanded for gene therapy approaches. Finally, we tested the potential of this screening system in other versions of *piggyBac* transposase.

## 1. Introduction

DNA transposons are mobile genetic elements present in the genomes of all organisms that can mobilize from one location to another by a “cut and paste’’ mechanism. Since the discovery of the first DNA transposon in maize in 1950 [1], these elements have been widely used as attractive tools for genetics and functional genomics in a wide range of organisms, including plants [2], insects [3], *Caenorhabditis* [4], fungi, and crustaceans [5]. However, their application in mammalian cells, especially in human cells, has been hampered due to the lack of or low transposition efficiency of the transposase. It is well known that the DNA transposon represents an attractive tool for gene delivery to achieve a long-term gene expression in immunotherapy applications [6], because of its low potential genotoxicity and safety risk of viral vectors for genetic modification in human cells [7,8,9]. Thus, it is urgently needed to find a high-efficiency transposase that can promote the development of non-viral vehicles for gene therapy.

Among a number of DNA transposons from different families, *piggyBac* demonstrates higher transposase flexibility and transposition activity than *Sleeping Beauty*, *Tol2*, and *Mos1* in mammalian cells [10]. The *piggyBac* transposon system was firstly isolated from the cabbage looper moth, *Trichoplusia ni*, of the order Lepidoptera while propagating Baculo-virus in the TN-386 cell line in the 1980s [11]. The *piggyBac* superfamily shows precise excision and integration at TTAA sites and leaves no trace of transposon [12]. Furthermore, it has many desirable features, including seamless excision, limited viral genotoxicity [8], a large cargo capacity (up to 150 kb) [13], and a highly engineered structure [14]. The transgene mediated by the *piggyBac* transposon system has several advantages that make it flexible compared to other transposon systems, including its high integration efficiency [15], its stable integration, its long-term expression, its single copy integration [16], its insertion site being easy to locate, and its easy manipulation [17]. Taking advantage of these unique characteristics, it has been widely used in the field of gene discovery via insertional mutagenesis, generation of transgenic animals, engineering stable cell lines, modification of clinically relevant cells (iPSCs, HSCs, T cells) [18], and gene transfer in vivo [19].

Even though there are thousands of DNA transposons systems, the vast majority of them are composed of a transposon and a transposase [17]. The transposon, flanked by two terminal inverted repeats (IRs), is usually the target gene cassette for transposition, whereas the transposase, through a “cut and paste” mechanism, recognizes the IRs, excises the transposon, and integrates it in another site of the genome. Therefore, engineering a high activity transposase is significant to increase the transposition efficiency for the application of gene editing in mammalian cells. Many optimized transposases have been isolated and extensively applied for gene editing in mammals. In 2009, Sleeping Beauty transposase (SB100X) was successfully isolated and showed a 100-fold enhancement of efficiency compared to the first-generation transposase [20]. In 2019, Querques et al. used a rational protein design based on the crystal structure of the hyperactive SB100X variant to create hsSB with enhanced solubility and stability [21]. For *piggyBac,* Yusa et al. found that a mammalian codon-optimized PBase mediates more efficient transposition than the original version [14]. Later, they generated a new hyperactive *piggyBac* transposase (hyPBase) that exhibits a marked hyperactivity through combining all mutations of five hyperactive mutants [22]. This research greatly expanded the utility of the *piggyBac* transposon for applications in mammalian genetics and gene therapy.

In the current study, we reported a simple effective screening system and a high-throughput screening process in yeast. Using this screening system, we isolated 15 hyperactive *piggyBac* transposases that exhibited higher transpose activity compared to hyPBase in yeast and four mutants that exhibited higher transpose activity in mammalian cells from 3000 hyPBase mutants. Finally, the most hyperactive transposase, bz-hyPBase, was verified and showed the potential to improve the editing efficiency of T cells.

## 2. Results

### 2.1. Construction of an Efficient Screening System in Yeast

Several reports have demonstrated that there is a requirement for a TTAA nucleotide sequence for *piggyBac* integration [12]. We used a transposon-containing resistance gene *kanMX*4 between 5′IR and 3′IR to interrupt the *ura*3 gene in plasmid pRS316 at the “TTAA” point. The transposase containing the wild-type PBase (WT PBase) gene under the galactose-inducible control of the GALS promoter was inserted into the same plasmid at the multiple cloning site to generate the reporting plasmid pRS-URA3-WT PBase. The schematic is shown in Figure 1A and the map of the plasmid is shown in Appendix A. The reporting plasmid containing a transposon element and a transposase element was transformed into *Saccharomyces cerevisiae* (*S. cerevisiae*) ura^−^ strain BJ2168 to form a screening reporting system. Due to the presence of the transposon, the *ura3* gene cannot express normally, and thus the strain is a uracil auxotroph. The strain can revert to uracil prototrophy when the transposon is removed from the plasmid. Thus, detecting the frequency of clones that reverse the growth on the -ura medium could reflect the activity of excision. Measuring the frequency of clones that grow in YPD medium with geneticin (G418) resistance can reflect the activity of integration.

To verify the effectiveness of this screening system, we transformed the plasmid pRS-URA3-WT PBase and the plasmid pRS-URA3-control without transposase into yeast ura^−^ strain BJ2168 separately. As shown in Figure 1C,D, after 24 h of induction, the strains that transferred into transposase could revert to uracil prototrophy and formed clones on the -ura medium plate or grew in the -ura liquid medium. Meanwhile, strains containing only the transposon could not reverse the growth in -ura medium. These findings indicate that an efficient one-plasmid screening system in a yeast ura^−^ strain has been established. Screening on an agar plate could be used to screen a more active or a less active mutant from any transposases, and the screening in liquid media could be used to screen active mutants from inactive transposases.

### 2.2. Improvement of a High-throughput Screening Process

We generated a mutant library of *piggyBac* transposase DNA (optimized hyPBase as an example) by error-prone PCR using a Clonthech Diversify PCR Random Mutagenesis Kit, and the mixed mutant DNA was inserted into the screening reporting vector by replacing the original transposase as described in the Materials and Methods section. Theoretically, the storage capacity of the mutant library is huge, as the *piggyBac* transposase is 1800 bp and the mutations can vary between 1 and 9 per 1000 bp.

To screen in a thorough and efficient way, the screening process was improved in two steps, as shown in Figure 2. In the first qualitative step, the mutants that gave rise to increased growth of ura^+^ revertants compared to non-mutation transposase were isolated for further analysis. In the second quantitative step, the concentration of the mutants (Figure 2F,G) from the first step was adjusted to keep their growth consistent. The transposition efficiency was calculated by ratio of the transposed clones and the total clones. Finally, the isolated clones that exhibited high transpose activity were picked out for mutation site analysis. By applying this high-throughput screening process, we could simultaneously screen 1000 mutant transformants by using a two-step screening process.

### 2.3. Isolation of Hyperactive piggyBac Transposase Mutants

As mentioned previously, hyPBase—obtained by Querques and her colleagues—exhibited much higher transposition activity than WT PBase that contains seven amino acid mutations (I30V, S103P, G165S, M282V, S509G, N538K, and N571S). To obtain more efficient transposase for mammalian cells, especially in human cells, we used optimized hyPBase derived from hyPBase using human codon preference as a template for mutation. The codon-optimized sequence is reported in Appendix A. In the preliminary screen, we examined 3000 transformants and obtained 49 mutants for further quantitative analysis. Consequently, we isolated 15 hyperactive mutants, and a 2–17-fold increase was found in transpose activity in *S. cerevisiae* (Figure 3A). The mutation sites of these 15 candidates distributed over the whole coding sequence and the number of mutation sites was from 1 to 8. The exact mutations of the candidates are reported in Appendix A.

Here, we chose the h62-16 mutant (denoted as: bz-hyPBase), which showed the highest activity in human cells to repeat the transposition assay in yeast. As shown in Figure 3B, hyPBase containing seven amino acid mutations exhibited higher transpose activity than WT PBase. The optimized hyPBase showed higher transpose activity than hyPBase. Moreover, our obtained bz-hyPBase displayed higher transpose activity than hyPBase and the optimized hyPBase. Figure 3B represented the clones’ reversed growth on -ura medium plate, and Figure 3C,D showed the statistical results. By sequencing, we obtained the following four mutation sites of bz-hyPBase is I82N (ATC→AAC), V109A (GTG→GCG), K290K (AAG→AAA), and Q591R (CAG→CGG). The transposition assay proved that our transposase screening system in yeast could be an effective and repeatable strategy to isolate target mutants from numerous transformants.

### 2.4. Hyperactive Mutants in Mammalian Cells

To investigate whether the mutants isolated in yeast could also display an elevated transpose activity in mammalian cells, the coding sequence of each mutant was transferred from the yeast expression vector to a mammalian expression vector with an EF-1α core promoter to generate the plasmids pLoxP-optimized hyPBase and pLoxP-mutant expressing the transposase. The transposon with the EGFP gene was cloned into the vector pSAD-EGFP expressing green fluorescent protein. A map of the plasmids is shown in Appendix A. The two plasmids expressing the transposase and the transposon were coelectrotransformed into Chinese hamster ovarian carcinoma (CHO) cells. The transposon with EGFP was inserted into the genome under the action of the transposase to stably express the green fluorescent protein. After 7 days of culture, the cells that expressed EGFP were counted by flow cytometry. The results show that among the 15 candidates, only four mutants showed higher transpose activity than non-mutated transposase (Figure 4A). The mutant h34-22 with the highest transposition efficiency in yeast did not exhibit the highest transposition efficiency in CHO cells. The highest transpose activity mutant in CHO cells was the mutant h62-16 (denoted as bz-hyPBase). It exhibited 2.88-fold higher increase in transposition efficiency in yeast and a 1.62-fold higher increase in cells compared to non-mutated transposase.

To define the transpose activity of bz-hyPBase quantitatively, we transformed bz-hyPBase into CHO cells again and compared the transpose activity with several controls. As shown in Figure 4B, the transpose activity of hyPBase, optimized hyPBase, and bz-hyPBase in CHO cells showed a similar increasing tendency in yeast. This finding partially demonstrates that the yeast screening model could be suitable to obtain a high transposase mutant in mammalian cells. The results also show that 72.2% of cells transfected with bz-hyPBase were able to transpose the EGFP gene to the genome to stably express green fluorescence. Meanwhile, only 49.6% of cells transfected with the optimized hyPBase could complete gene transposition. Furthermore, the control cells containing plasmid pSAD-EGFP showed no EGFP expression. In conclusion, the presented results demonstrate that this screening system in *S. cerevisiae* could be an efficient tool to obtain mutants exhibiting much higher transpose activity in mammalian cells.

### 2.5. bz-hyPBase Improved the Editing Efficiency of T cells

Previous findings have demonstrated the traceless excision and the large gene cargo capacity property of the *piggyBac* transposon, which enabled us to generate transgene-free cells and transgene-free cells animals while maintaining an unaltered genome. It is well known that the *PiggyBac* transposon system is an important non-viral transfection system for CAR-T gene delivery.

To investigate whether the mutant that shown higher activity in CHO cells could increase the editing efficiency in T cells, we measured the transpose activity of h31-11, h33-14, h61-5, and h62-16 in human T cells. A two-plasmid transposon system, in which one plasmid can express the transposon (pSAD-EGFP) and the other plasmid (pLoxP-optimized hyPBase/pLoxP-mutant) can express transposase, was used to measure the transpose activity. Under the action of the transposase, the transposon was inserted into the genome to edit the T cells. Then, the positive cells were measured by flow cytometry. Among the four mutants, h33-14 and h62-16 exhibited higher transpose activity than the optimized hyPBase (Appendix A). In order to eliminate the different quality of T cells from different persons, we chose three donors to test the positive rate of EGFP after 7 and 14 days. The transpose activity of h33-14 was varied due to the different donors (Appendix A), while h62-16 (bz-hyPBase) exhibited much higher stable transpose activity. As shown in Figure 5, the editing efficiency of T cells with bz-hyPBase was increased by 1.3–2-fold compared to cells with the optimized hyPBase. Meanwhile, the control cells containing the transposon plasmid pSAD-EGFP could not complete gene transposition. As a result, the number of EGFP-positive colonies was significantly increased by using bz-hyPBase compared to the optimized hyPBase, indicating that the bz-hyPBase could enhance the editing efficiency of T cells.

### 2.6. The Effectiveness of the Screening System

To verify the effectiveness of the screening system, we optimized wild-type *piggyBac* transposase by optimized codon content for human cells. Then, 15,000 transformants from wild-type *piggyBac* transposase were screened by using this efficient screening system described above. We obtained six mutants that exhibited higher transpose activity than optimized WT PBase in human T cells (Appendix A). As our mutant library was not saturated, more hyperactive mutants might be identified by our screening system in the future. Therefore, the screening system is suitable for the other version of *piggyBac* transposase.

Furthermore, we integrated all mutation sites into one sequence, and measured the transposition efficiency of this sequence in CHO cells. For optimized hyPBase mutants, we integrated the mutations of h33-14 and h62-16 into one sequence to generate mutant co-1614. For optimized WT PBase, we integrated the mutations of op12-27 and op12-56 into one sequence to generate mutant co-2756. As shown in Appendix A, both co-1614 and co-2756 exhibited lower transpose activity than the highest mutants. This might be due to the influence of mutations on the structure of the transposase.

## 3. Discussion

It is well known that DNA transposons have been widely used for transgenesis and insertional mutagenesis in the mammalian genome. The *piggyBac* transposon system, which is considered the most attractive site-specific non-viral vector, demonstrated high efficiency in the genomic engineering of mammalian cells for preclinical applications. However, further improvements in their transpose activity are required for industrialization and clinical applications.

In this study, we established a one-plasmid effective screening system in *S. cerevisiae* and isolated a hyperactive *piggyBac* transposase for mammalian cell applications. Since yeast is one of the simplest eukaryotic organisms and has been investigated extensively, it was selected as a model organism in our high-throughput screening system. Firstly, we inserted the transposon into the *ura3* gene at the TTAA site and combined the inducible transposase into one plasmid to generate the screening report plasmid. The plasmid was transformed into a yeast ura^−^ strain to generate the screening system. Secondly, we generated a mutant library of hyPBase by error-prone PCR and screened it by a two-step screening process. The candidates were subsequently verified in mammalian cells, and 4 out of 3000 hyPBase mutants exhibited higher transpose activity. Finally, the ability of most hyperactive transposase bz-hyPBase to improve the editing efficiency of T cells was also verified. Besides, we tested the potential of this screening system in other versions of *piggyBac* transposase.

It has been reported that *piggyBac* transposition is a host factor-independent reaction, since the transposition could be reconstituted in vitro by using purified PBase and DNA elements [23]. However, only one fourth of the mutants screened in yeast exhibited higher transpose activity in our study. This observation might be due to several reasons. The usage frequency of amino acids is different between yeast and mammals; the increased transposition efficiency in yeast might be due to increased amino acid usage of transposase. On the other hand, the reaction temperature could affect the catalytic activity of transposase [24], since yeast grows at 30 °C while mammalian cells grow at 37 °C. Besides, epigenetic modifications can also influence the transposition frequency in a species-specific manner [25,26].

The non-viral *piggyBac* transposon system has greatly widened our understanding of DNA transposons and has opened new areas of research with clinical implications, especially in cell therapy. *piggyBac* has been used to generate mouse- [27] and human-induced pluripotent stem cells [28] (hiPSCs), and to modify hiPSCs, human embryonic stem cells (hESCs), human hematopoietic stem cells (HSCs) [29], and human T lymphocytes [30]. The use of *piggyBac* to efficiently modify human T cells for expression of chimeric antigen receptor demonstrated functional activity to kill antigen-specific tumor [31]. More and more human CAR-T clinical trials using *piggyBac* transposon have been approved, since it is considered more secure and lower cost than the widely used viral vectors [32]. Newer hyperactive *piggyBac* elements containing more hyperactive transposase and higher specific transposon are urgently needed. In addition to the increasing transpose activity, improving the targeting of *piggyBac* elements to user-defined chromosomal locations is also important to improve the safety and specificity in gene transfer approaches.

## 4. Materials and Methods

### 4.1. Strains, Growth Condition, and Cell Culture

*Saccharomyces cerevisiae* BJ2168 (*MATα prc*l-407 *prb*1-1122 *pep*4-3 *leu*2 *trp*1 *ura*3-52 *gal*2) was purchased from American Type Culture Collection (ATCC, Manassas, VA, USA). *S. cerevisiae* and its derivative strains were grown in yeast peptone dextrose (YPD) medium at 30 °C and shaking at 220 rpm. The media were supplemented with appropriate antibiotics *Escherichia coli*, 100 μg/mL ampicillin and *S. cerevisiae*, 150 μg/mL (G418) to maintain the plasmids when necessary.

CHO (Chinese hamster ovarian carcinoma) cells were obtained from ATCC and grown in Dulbecco’s modified eagle medium (DMEM; Corning, Corning, NY, USA) supplemented with 10% fetal calf serum (FBS, Gibco, USA). The human peripheral blood mononuclear cells (PBMC) of healthy donors were purchased from AllCells (AllCells, Silicon Valley, CA, USA) under a protocol approved by the Ethics Committee of Chinese People’s Liberation Army General Hospital (S2019-063-01), and isolated from heparinized blood using Ficoll-Paque density gradient centrifugation. PBMCs were plated in serum free AIM-V media (Life Technologies, Grand Island, NY, USA) and allowed to adhere to 0.22 μm filter-capped culture flasks (Corning, Corning, NY, USA). After 4–6 h, the non-adherent cells were collected as T cells, which were transiently cultured in AIM-V medium supplemented with 2% FBS (Gibco, Grand Island, NY, USA) before electroporation. CHO cells and PBMC cells were maintained at 37 °C in a humidified atmosphere containing 5% CO_2_.

### 4.2. Plasmid Construction

To construct the reporting vectors pRS-URA3-PB in yeast, the transposon was cloned into the vector pRS316 at the TTAA point of the *ura3* gene by homologous recombination. The transposase with an inducible promoter was cloned into the vector at a polyclonal digestion site. The specific operations were as follows: (1) the vector fragment was amplified from pRS316 using primers pURA-F and pURA-R. (2) The transposon fragment with a homology arm on vector pRS316 was amplified from transposon kanMX4-IR using primers pURA-IR-F and pURA-IR-R. The transposon kanMX4-IR containing the resistance gene *kanMX4* between 5’IR and 3’IR were synthesized by GENEWIZ. (3) The vector fragment and the transposon fragment were ligated using NEBuilder for recombination to generate the pRS-URA3-control plasmid. (4) The transposases (WT PBase, optimized hyPBase, optimized WT PBase) with an inducible GALS promoter were synthesized by the GENEWIZ company and cloned into the vector pRS-URA3-control using SacI and EcoRI to generate the pRS-URA3-WT PBase plasmid. Under the induction of 2% galactose, the GALS promoter could turn on the expression of transposase [33]. The sequence of transposases and the GALS promoter are listed in Appendix A. The primer sequence is listed in Table 1.

To construct the reporting vectors in human cells, the PBase element with an EF-1-α promotor and SV40 ployA were cloned into pLoxP to generate pLoxP-optimized hyPBase or a pLoxP mutant [34]. The transposon expressing the enhanced green fluorescent protein (EGFP) was cloned into pS338C to generate the pSAD-EGFP plasmid [34,35].

### 4.3. Mutant Library Construction

To generate a mutant library of optimized hyPBase, the mutated transposase fragments containing a 50–80 bp homology arm on vector were amplified from the vector pRS-URA3-optimized hyPBase with primer GR-F/GR-R using a Clontech diversify PCR Random Mutagenesis Kit (Clontech 630703) according to the manufacturer’s protocol. The mutations can vary between 1 and 9 per 1000 bp depending on different buffer conditions. They could also be accumulated by purification with the PCR products as a template. The screening report vector pRS-URA3-optimized hyPBase was linearized using XbaI and EcoRI. The mutated transposase fragments and the linearized vector were transformed into a yeast strain at a molar ratio of 10:1. The clones in which the mutated transposase integrated into the vector by homologous recombination were grown on YPD agar plates with G418 resistance for 2 days. To generate a mutant library of optimized WT PBase, error-prone PCR was performed with the primer GR-F1/GR-R1.

### 4.4. Screening of Yeast Mutants

The yeast strain, reporting plasmid, and the mutant library were described previously. The transformed single clones were firstly grown into the YPD medium supplemented with G418 resistance in a 96-well plate, and then transferred to a new 96-well plate with YPD medium containing 2% galactose to express transposase. After 24 h of grown, the yeast solution was diluted to the appropriate concentration (10^−2^ and 10^−3^) and 10 μL of the solution was taken to spot onto plates lacking uracil. According to our experiences, a maximum of 16 samples (4 × 4) can be spotted per plate to prevent cross-pollution. Mutants that gave rise to increased numbers of ura^+^ revertants were further analyzed quantitatively.

The obtained mutant clones were grown in tubes for 24 h and transferred to YPD medium supplemented with 2% galactose with the same amount of yeast. After 24 h of grown, the yeast solution was diluted to the appropriate concentration and 10 μL of the solution was taken to spread on the -ura medium plate (the optimum decimal dilutions were 10^−2^ and 10^−3^) to count the transposed clones, and the YPD plate (the optimum decimal dilutions were 10^−4^ and 10^−5^) to count the total clones as control for 48 h. The transpose activity efficiency = number of clones transposed/total number of clones = (number of clones in -ura medium × dilution factor)/(number of clones in YPD medium × dilution factor) × 100%. The transposition efficiency was calculated and compared with the non-mutation one.

The mutants that exhibited higher transpose activity were picked out for mutation analysis. Yeast plasmid DNAs were isolated using the Zymoprep Yeast Plasmid Miniprep I kit. The mutant transposase fragments were sequenced by GENEWIZ for mutation site analysis.

### 4.5. Transposition Assay in Mammalian Cells

To determine the transposition efficiency in CHO cells, 1 × 10^7^ cells were coelectroporated with 4 μg of each transposase expression vector pLoxP-optimized hyPBase/pLoxP-mutant and transposon expression vector pSAD-EGFP using an Amaxa^®^ Human T Cell Nucleofector^®^ Kit by utilizing an electroporator (Lonza, Basel, Switzerland) according to the manufacturer’s protocol. Electroporated cells were cultured in DMEM medium for 14 days. Cell culture medium was changed every 2 days.

To determine the transposition efficiency in T cells, 5 × 10^6^ cells were coelectroporated with 4 μg of each transposase expression vector pLoxP-optimized hyPBase/pLoxP-mutant and transposon expression vector pSAD-EGFP as described above. The transfected T cells were specifically stimulated with anti-CD3/CD28-antibody for 5 days in 2% FBS-AIM-V medium supplemented with 200 U/mL recombinant human interleukin-2 (IL-2). Then, the activated cells were cultured in 2% FBS-AIM-V medium containing 100 U/mL IL-2 for 14 days. Cell culture medium was changed every 3 days.

### 4.6. Flow Cytometry

Cells were collected after 7/14 days of culture and washed twice in PBS. Then, 1 × 10^6^ cells were resuspend in 400 μL PBS for flow cytometry. Cells that expressed GFP (green fluorescence protein) were detected by a Beckman Coulter Gallios™ flow cytometer using Channel FITC. The data were analyzed with FlowJo vX.0.7 software (Beckman Coulter Inc., Indianapolis, IN, USA).

### 4.7. Statistical Analysis

Statistical analyses were carried out using GraphPad Prism 5.0 (GraphPad Software, San Diego, CA). Data were analyzed using unpaired *t*-tests to compare two different conditions and analysis of variance for more conditions. All error bars show the standard errors of the mean (SEM). Statistically significant differences were considered as follows: *p* ≥ 0.05 (ns), *p* < 0.05 (*), *p* < 0.01 (**), *p* < 0.001 (***), and *p* < 0.0001 (****).

## Figures and Tables

**Figure 1 ijms-21-03064-f001:**
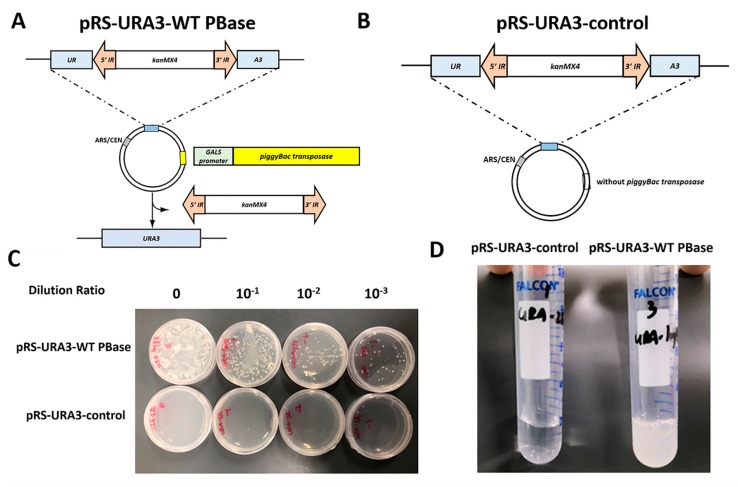
Efficient screening system in *Saccharomyces cerevisiae*. (**A**) Schematic representation of the screening reporting plasmid pRS-URA3-PB. The *ura3* gene was separated by the *piggyBac* transposon at the TTAA site, which completely disturbed the normal expression. The *piggyBac* transposase with a galactose-inducible GALS promoter was inserted into the multiple cloning site. The transposon could be excised by the transposase under galactose induction, recovering the expression of the *ura3* gene. (**B**) Schematic representation of control plasmid pRS-URA3-control. (**C**) The plasmid pRS-URA3-WT PBase and the plasmid pRS-URA3-control were transformed into a yeast ura^−^ strain. After 24 h post-induction, the strains that had transferred into transposase could revert to uracil prototrophy, while the control strains could not. (**D**) The transposition assay in liquid media. IR, inverted repeat.

**Figure 2 ijms-21-03064-f002:**
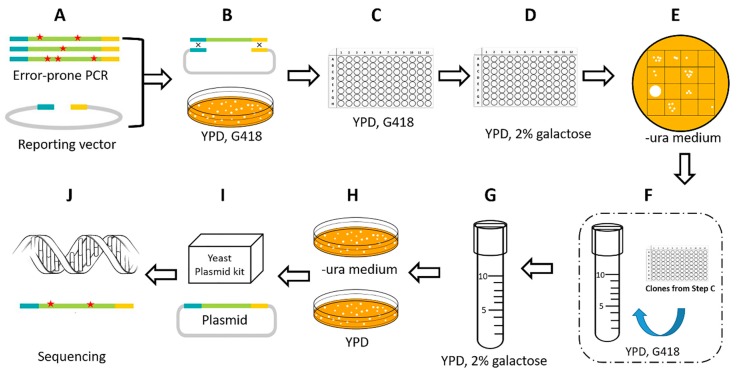
High-throughput screening process. (**A**,**B**) Generation of a mutant library of *piggyBac* transposase by error-prone PCR and a gap-repair system in yeast. (**C**–**E**) The first qualitative screening step. The mutants were grown and induced in a 96-well plates. After 24 h of galactose induction, the solution was diluted and spotted onto the uracil-deficient medium. Mutants that could reverse more clones were chosen for further quantitative analysis. (**F**–**H**) The second quantitative screening step. The mutants that were chosen from E and isolated from the C were grown and induced in 12 mL tubes. The concentration was adjusted to ensure a consistent growth condition. After 24 h of galactose induction, the solution was diluted and spread on the -ura medium plate and the YPD plate as control. The transposition efficiency was calculated and compared to the non-mutation one. (**I**) The isolated clones that exhibited high transpose activity were picked out and the plasmids were extracted. (**J**) The transposase fragment was sequenced for mutation site analysis.

**Figure 3 ijms-21-03064-f003:**
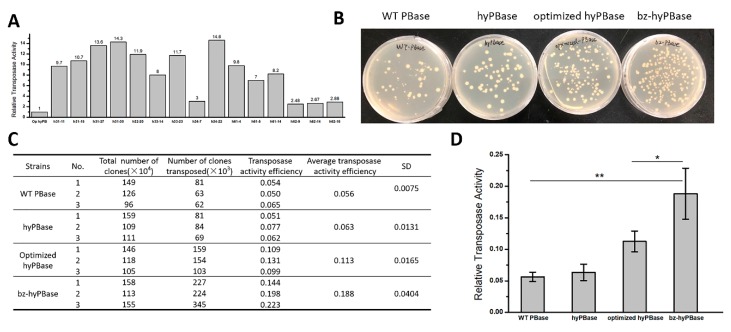
Isolation of *piggyBac* transposase mutants in *S. cerevisiae.* (**A**) Relative activities of 15 hyperactive mutants in yeast. The transposition efficiency of optimized hyPBase was set as 1.0, and the increase folds of hyperactive mutants were marked at the top of the column. (**B**) The transposition assay of wild-type (WT) PBase, hyPBase containing seven amino acid mutations, the codon-optimized hyPBase, and bz-hyPBase. bz-hyPBase, which we isolated with four mutation sites, exhibited significantly higher transposition efficiency than others. (**C**) The transposition efficiency of WT PBase, hyPBase, optimized hyPBase, and bz-hyPBase. The transposition efficiency = number of clones reversed growth on -ura medium plate/total number of clones × 100%. (**D**) Relative activities of WT PBase, hyPBase, optimized hyPBase, and bz-hyPBase. *, *p* < 0.05 and **, *p* < 0.01.

**Figure 4 ijms-21-03064-f004:**
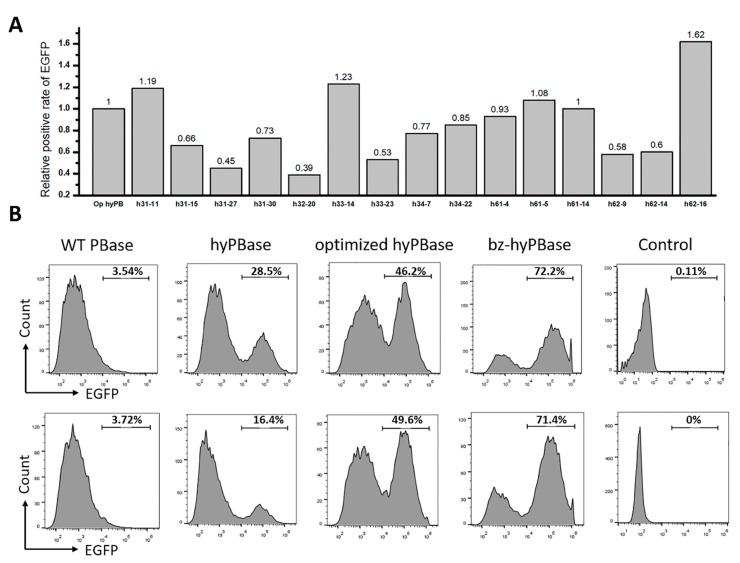
Hyperactive *piggyBac* transposase mutants in mammalian Cells. (**A**) Relative activities of 15 hyperactive mutants in Chinese hamster ovarian carcinoma (CHO) cells. The transposon plasmid pSAD-EGFP containing EGFP in transposon and the transposase plasmid pLoxP-optimized hyPBase or pLoxP -mutant containing transposase were coelectrotransformed into CHO cells. After 7 days of culture, the cells that expressed green fluorescent protein were counted by flow cytometry. (**B**) Relative activities of WT PBase, hyPBase, optimized hyPBase, and bz-hyPBase. The upper and the lower row are two repeated results. The control cells containing plasmid pSAD-EGFP only showed almost no green fluorescent protein expression. The X-axis represented the intensity of EGFP, and the Y-axis represented the number of cells. The values represented the percent of EGFP cells in each sample.

**Figure 5 ijms-21-03064-f005:**
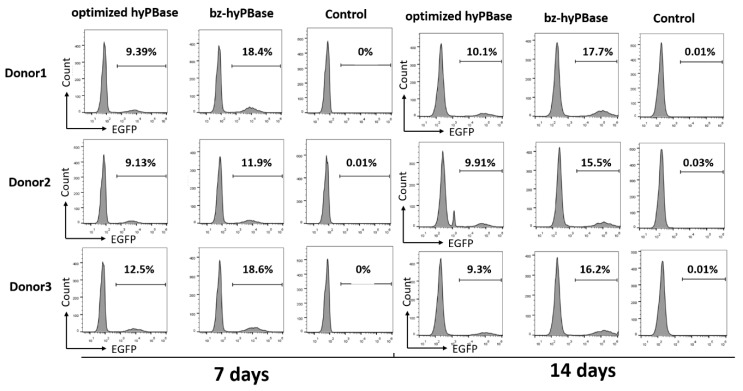
bz-hyPBase improved the editing efficiency in T cells. The two-plasmid transposon system (pLoxP-optimized hyPBase/pLoxP-mutant to express transposase, and pSAD-EGFP to express transposon) were coelectrotransformed into human T cells. After 7 and 14 days of culture, positive cells that completed gene transposition were measured by flow cytometry. On the right were the control cells containing the transposon plasmid pSAD-EGFP only. The X-axis represented the intensity of EGFP, and the Y-axis represented the number of cells. The values represented the percent of EGFP cells in each sample. The experiment was repeated by three groups T cells from three different donors.

**Table 1 ijms-21-03064-t001:** Primer sequence.

**Primer**	**Primer Oligonucleotide (5′–3′)**	**Application**
pURA-F	AAGCCGCTAAAGGCATTATCCGCC	Plasmid linearization
pURA-R	AACTGTGCCCTCCATGGAAAAATCAGTC	Plasmid linearization
pURA-IR-F	GACTGATTTTTCCATGGAGGGCACAGTTAACCCTAGAAAGATAGTCTGCGTAAAATTGACGCATGCGAC	Transposon insertion
pURA-IR-R	GGCGGATAATGCCTTTAGCGGCTTAACCCTAGAAAGATAATCATATTGTG	Transposon insertion
GR-F	TAATCAGCGAAGCGATGA	Error-prone PCR
GR-R	CAGCATGCCTGCTATTGTCTTCC	Error-prone PCR
GR-F1	CCACTTTAACTAATACTTTC	Error-prone PCR
GR-R1	CCCTCACTAAAGGGAACAAAAGCTG	Error-prone PCR

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
