# Peer review of "An efficient Screening System in Yeast to Select a Hyperactive piggyBac Transposase for Mammalian Applications"

_ijms, 2020, doi:10.3390/ijms21093064_

Round 1
Reviewer 1 Report
In the manuscript “An efficient screening system in yeast to select a hyperactive piggyBac transposase for mammalian applications,” the authors used a practical yeast screening system to identify a piggyBac derivative that would be highly active in mammalian cells. Nevertheless, I am compelled to advise rejection of this manuscript for the reasons state below.
In the following three paragraphs, I try to summarise my understanding of what the authors did, despite the manuscript being written in an unacceptably mish-mash and convoluted way.
> Summary
The authors constructed plasmid pRS-URA-control and inserted randomly-mutagenised versions of piggyBac transposase into it. First, they took the wild type piggyBac transposase and mutagenized it. Among 15,000 analysed yeast transformants, six expressed a more active transposase. The authors tried to combine mutations from all six of these transposases into one gene. However, the resulting transposase showed unexpectedly low activity in CHO cells.
Next, the authors took previously published hyperactive hyPBase transposase, optimised its codon content for human cells, and then mutagenized it. Among 4,000 analysed yeast transformants, 15 expressed transposases that were hyperactive in yeast cells. However, only four of those transposases were more active in CHO cells than codon-optimised hyPBase. Once again, the authors tried to combine mutations from all four of these transposases into one gene, but the resulting transposase had unexpectedly low activity. Interestingly, high transposition frequency in yeast did not predict high transposition frequency in CHO cells.
Finally, the authors transfected three lines of T cells with the mutant transposase that showed the highest activity in CHO cells. This transposase transposed DNA from donor plasmid into the chromosomal DNA better than previously developed piggyBac versions.
> Obtained results
The submitted work has some scientific merit, but not in the form presented here. An experimental system is just a tool, not the pinnacle of scientific work, especially when it consists of only one plasmid. Novel, highly active transposase variants are worth reporting about, but they should be well characterised. The authors should report exact mutations for each of the transposase variants, both for those that are more active in mammalian cells and those that are more active in yeast cells. If mutagenesis is performed on a codon-optimised sequence, the sequence itself should be reported as well, either in the supplementary materials or in Genbank. In the present manuscript, the authors report base substitutions for only one variant, even though results clearly show that its higher activity partly stems from codon optimisation. Transposase activity should be measured in triplicate for all promising variants, not for only one of them.
> Writing
The manuscript is written in a very convoluted way. Large sections of the Results (most notably, parts of 2.1. and 2.2.) should be moved into Materials and Methods. Breaking all norms, parts of the Discussion introduce completely new, never-before-mentioned results. The methodology for these new results in Discussion is omitted from the Materials and Methods. Finally, the manuscript focuses on the “efficient screening system in yeast for mammalian applications” for which the authors themselves admit that “the increased tendency [of transposase activity in CHO cells] had no correlation with that in yeast”. I would suggest focusing on newly-developed hyperactive transposase variants.
Below is the list of several other specific concerns.
1) The manuscript has severe grammatical problems: incomplete sentences, frequent misspellings (e.g. flod instead of fold)... However, a more severe problem is the incorrect use of terminology. By transforming yeast cells, one does not obtain transformers but transformants. Yeast cells are not activated; they are grown. Cell culture is the term that should be used when referring to the culture of mammalian cells, not when referring to the yeast cells. Abbreviation for Saccharomyces cerevisiae is S. cerevisiae, not S. Cerevisiae. Gene does not encode resisitance but resistance. It is very unusual to say that the transposase transports DNA; rather, it transposes it. One does not use URA- medium but -ura medium (or medium without uracil). Yeast phenotype is not URA3- but ura- (with - in the superscript). Yeast genotype is not URA3- but ura3. Do authors, instead of dilution gradients mean decimal dilutions? The complete genotype of strain BJ2168 should be given. Detailed protocol for induction of GALS promoter was omitted.
2) In Figure 1A and 1B:
- There is a blank space between transposon’s inverted repeats and the first/last fragment of the URA3 gene. Thus, the scheme suggests that transposition includes degradation of the sequence around the transposon, which is incorrect. There should not be any space between labels URA and IR, nor between labels IR and A3.
- Reconstituted URA3 gene is not written in italic, as the convention dictates.
- It is unclear what label “ura” above the plasmids mean; did the authors intend to write ura3?
- The plasmid shown is not pRS316, but its derivative pRS-URA-PBase. Thus, the label pRS316 inside the plasmid is confusing and should be removed from the Figure. Moreover, pRS316 contains the S. cerevisiae ARS/CEN region, which has important implications for the constructed experimental system. As such, it should be clearly labelled.
- Does the label G418 refer to the antibiotic used to maintain the experimental system in the transformants? If so, the reader could infer antibiotic used from the name of the resistance gene. Or is “G418” name of the resistance gene? That choice would be confusing as both the antibiotic and the gene would have the same name. Thus, neither is appropriate. Additionally, the paper does not specify which resistance gene was actually used, only that it was synthesised de novo. Instead of using labels “resistance gene” and “G418”, the authors should clearly state which sequence was used to provide resistance, e.g. was it often used kanMX4?
- Conventionally, the plasmid name starts with small letter p, which denotes that the molecule is indeed a plasmid. However, in the figures and manuscript as a whole, the authors interchangeably use PRS and pRS.
In Figure 1D, labels appear to be switched. But actually, the whole Figure 1D is unnecessary. Cells growing on the solid medium can grow in liquid medium as well.
3) Figure 2 suggests that for the “quantitative” screening of mutants, YPD supplemented with G418 in Fig. 2F was inoculated with patches of cells grown on -ura medium (Fig. 2E). Was this indeed the case? As the experimental system was based on the centromeric plasmid, which can be present in more than one copy per cell, it is possible that in some yeast cells one plasmid molecule underwent transposition while the other plasmid molecule did not. These cells would have phenotype ura+ G418R and could grow in YPD supplemented with G418. Such cells would in later steps falsely appear to have a very active transposase. Alternatively, did the authors instead first identify interesting strains in Fig. 2E and then used cells grown in Fig. 2C to inoculate Fig. 2F? This course of action would prevent unwanted enrichment of ura+ cells. If so, this sequence of steps should be denoted clearly.
4) It is unclear which version of piggyBac was used in which mutagenesis protocol. Section 2.1. mentions “PBase gene”. Section 2.2. mentions “a mutant library of piggyBac transposase DNA”. Section 2.3. mentions “optimized-hyPBase as a template for mutation”.
5) The plasmid maps of ploxP-PB, ploxP-PB-mutant, and pSAD-EGFP should be included in the manuscript, as are the plasmid maps of Fig. 1A and Fig. 1B. References for the plasmids used in the construction of all newly-constructed plasmids should be present in the manuscript. It is unclear what is in the upper and what in the lower row in Fig. 4B.
Author Response
Response to Reviewer 1 Comments
In the manuscript “An efficient screening system in yeast to select a hyperactive piggyBac transposase for mammalian applications,” the authors used a practical yeast screening system to identify a piggyBac derivative that would be highly active in mammalian cells. Nevertheless, I am compelled to advise rejection of this manuscript for the reasons state below.
In the following three paragraphs, I try to summarise my understanding of what the authors did, despite the manuscript being written in an unacceptably mish-mash and convoluted way.
Point 1:
> Summary
The authors constructed plasmid pRS-URA-control and inserted randomly-mutagenised versions of piggyBac transposase into it. First, they took the wild type piggyBac transposase and mutagenized it. Among 15,000 analysed yeast transformants, six expressed a more active transposase. The authors tried to combine mutations from all six of these transposases into one gene. However, the resulting transposase showed unexpectedly low activity in CHO cells.
Next, the authors took previously published hyperactive hyPBase transposase, optimised its codon content for human cells, and then mutagenized it. Among 4,000 analysed yeast transformants, 15 expressed transposases that were hyperactive in yeast cells. However, only four of those transposases were more active in CHO cells than codon-optimised hyPBase. Once again, the authors tried to combine mutations from all four of these transposases into one gene, but the resulting transposase had unexpectedly low activity. Interestingly, high transposition frequency in yeast did not predict high transposition frequency in CHO cells.
Finally, the authors transfected three lines of T cells with the mutant transposase that showed the highest activity in CHO cells. This transposase transposed DNA from donor plasmid into the chromosomal DNA better than previously developed piggyBac versions.
Response 1: We thank you for your great comments and feel sorry for making you convoluted in writing. We have used a professional English editing service that recommended by MDPI. Also, we will response the comments point-by-point.
Our aim is to provide researchers with a high-throughput screening method as a tool that can screen different versions of piggyBac transposases. First, we took the wild type piggyBac transposase to construct the screening system, but didn’t mutagenize it. The purpose of this step is to prove that the screening system is working. Next, we took previously published hyperactive hyPBase transposase to optimize its codon content for human cells, and then mutagenized it. The purpose of this step is to find the highest active transposase as possible by applying the system constructed above. Finally, we optimized wild type piggyBac transposase (optimized WT PBase) that optimised its codon content for human cells, and then mutagenized it. The purpose of this step is to prove that the system can applied in other versions of piggyBac transposase.
Point 2:
> Obtained results
The submitted work has some scientific merit, but not in the form presented here. An experimental system is just a tool, not the pinnacle of scientific work, especially when it consists of only one plasmid. Novel, highly active transposase variants are worth reporting about, but they should be well characterised. The authors should report exact mutations for each of the transposase variants, both for those that are more active in mammalian cells and those that are more active in yeast cells.
Response 2: Thanks for your suggestion. We have added the exact mutations in manuscript. Please see lanes 143. The exact mutations are reported in Table S2.
Point 3:
If mutagenesis is performed on a codon-optimised sequence, the sequence itself should be reported as well, either in the supplementary materials or in Genbank.
Response 3: Thanks for your suggestion. We have added the sequence of WT PBase, optimized hyPBase, optimized WT PBase in Table S1.
Point 4:
In the present manuscript, the authors report base substitutions for only one variant, even though results clearly show that its higher activity partly stems from codon optimisation. Transposase activity should be measured in triplicate for all promising variants, not for only one of them.
Response 4: Thanks for your comments. We chose all promising variants in CHO cells to measure its transposase activity in T cells actually, then chose the most promising variants h33-14 and h62-16 to measure for 3 donors. We have added this result in Figure S3 and Figure S4. Please see lane 207-222.
Point 5:
> Writing
The manuscript is written in a very convoluted way. Large sections of the Results (most notably, parts of 2.1. and 2.2.) should be moved into Materials and Methods.
Response 5: Thanks for your suggestion. We have revised the manuscript according to your comments in parts 2.1 (lane 71-84), parts 2.2 (lane 1075-120), legend of Figure 2 (lane 122-132), Materials and Methods (lane 332-352).
Point 6:
Breaking all norms, parts of the Discussion introduce completely new, never-before-mentioned results. The methodology for these new results in Discussion is omitted from the Materials and Methods.
Response 6: Thanks for your suggestion. We have moved this part to the Result 2.6. Please see lane 230-244. The methodology for optimized WT PBase is added in the Materials and Methods, too.
Point 7:
Finally, the manuscript focuses on the “efficient screening system in yeast for mammalian applications” for which the authors themselves admit that “the increased tendency [of transposase activity in CHO cells] had no correlation with that in yeast”. I would suggest focusing on newly-developed hyperactive transposase variants.
Response 7: Thanks for your comments. What I mean here is that the mutant with the highest transposition efficiency in yeast didn’t exhibit the highest transposition efficiency in CHO cells. According to experimental results, they have similar behaviour. I have modified my description here. Please see lane 177-179.
Point 8:
Below is the list of several other specific concerns.
- The manuscript has severe grammatical problems: incomplete sentences, frequent misspellings (e.g. flod instead of fold)... However, a more severe problem is the incorrect use of terminology. By transforming yeast cells, one does not obtain transformers but transformants. Yeast cells are not activated; they are grown. Cell culture is the term that should be used when referring to the culture of mammalian cells, not when referring to the yeast cells. Abbreviation for Saccharomyces cerevisiae is S. cerevisiae, not S. Cerevisiae. Gene does not encode resisitance but resistance. It is very unusual to say that the transposase transports DNA; rather, it transposes it. One does not use URA- medium but -ura medium (or medium without uracil). Yeast phenotype is not URA3- but ura- (with - in the superscript). Yeast genotype is not URA3- but ura3. Do authors, instead of dilution gradients mean decimal dilutions? The complete genotype of strain BJ2168 should be given. Detailed protocol for induction of GALS promoter was omitted.
Response 8: Thanks for pointing these out. We have used a professional English editing service that recommended by MDPI. We have modified the incorrectness as follows:
- Flod to fold.
- Transformers to transformants.
- Activated to grown.
- Cell culture is used when referring to mammalian cells only.
- Cerevisiae to S. cerevisiae.
- Transport activity to transpose activity.
- URA- medium to -ura medium. Yeast phenotype URA3- to ura-. Yeast genotype URA3- to ura3.
- Dilution gradients to decimal dilutions.
- The complete genotype of strain BJ2168 was added in lane 285.
- Detailed protocol for induction of GALS promoter was added in Materials and Methods. Please see lane 311-313.
Point 9:
2) In Figure 1A and 1B:
There is a blank space between transposon’s inverted repeats and the first/last fragment of the URA3 gene. Thus, the scheme suggests that transposition includes degradation of the sequence around the transposon, which is incorrect. There should not be any space between labels URA and IR, nor between labels IR and A3.
Response 9: Thanks for your comments. We have made the modification in Fig. 1 as you suggested.
Point 10:
Reconstituted URA3 gene is not written in italic, as the convention dictates.
It is unclear what label “ura” above the plasmids mean; did the authors intend to write ura3?
Response 10: Thanks, and this error has be revised. We have written URA3 gene in italic and deleted the label “ura” above the plasmid.
Point 11:
The plasmid shown is not pRS316, but its derivative pRS-URA-PBase. Thus, the label pRS316 inside the plasmid is confusing and should be removed from the Figure. Moreover, pRS316 contains the S. cerevisiae ARS/CEN region, which has important implications for the constructed experimental system. As such, it should be clearly labelled.
Response 11: Thanks for your comments. We have removed the label pRS316 inside the plasmid and labelled the ARS/CEN region as you suggested. The maps of pRS-URA3-WT PBase and pRS-URA3-control are reported in Figure S1.
Point 12:
Does the label G418 refer to the antibiotic used to maintain the experimental system in the transformants? If so, the reader could infer antibiotic used from the name of the resistance gene. Or is “G418” name of the resistance gene? That choice would be confusing as both the antibiotic and the gene would have the same name. Thus, neither is appropriate. Additionally, the paper does not specify which resistance gene was actually used, only that it was synthesised de novo. Instead of using labels “resistance gene” and “G418”, the authors should clearly state which sequence was used to provide resistance, e.g. was it often used kanMX4?
Response 12: Thanks for your great comments. G418 is the abbreviation of geneticin that is used to maintain the experimental system in the transformants, and kanMX4 is the resistance gene that can be used to provide resistance. We have modified all description in manuscript.
Point 13:
Conventionally, the plasmid name starts with small letter p, which denotes that the molecule is indeed a plasmid. However, in the figures and manuscript as a whole, the authors interchangeably use PRS and pRS.
Response 13: Thanks. We have modified this mistake.
Point 14:
In Figure 1D, labels appear to be switched. But actually, the whole Figure 1D is unnecessary. Cells growing on the solid medium can grow in liquid medium as well.
Response 14: Thanks for your comments. We have changed the labels in Figure 1D. The screening in liquid is used to prove that our screening system is very rigorous. If researchers want to screen active mutants from inactive transposases, screening in liquid is a simpler and more efficient method than plate screening. The mutants that could grow in -ura liquid media are the active transposases.
Point 15:
- Figure 2 suggests that for the “quantitative” screening of mutants, YPD supplemented with G418 in Fig. 2F was inoculated with patches of cells grown on -ura medium (Fig. 2E). Was this indeed the case? As the experimental system was based on the centromeric plasmid, which can be present in more than one copy per cell, it is possible that in some yeast cells one plasmid molecule underwent transposition while the other plasmid molecule did not. These cells would have phenotype ura+ G418R and could grow in YPD supplemented with G418. Such cells would in later steps falsely appear to have a very active transposase. Alternatively, did the authors instead first identify interesting strains in Fig. 2E and then used cells grown in Fig. 2C to inoculate Fig. 2F? This course of action would prevent unwanted enrichment of ura+ cells. If so, this sequence of steps should be denoted clearly.
Response 15: Thanks for your great comments. We used cells grown in Fig. 2C to inoculate in Fig. 2F. We have modified Figure 2 and explained in the screening process in figure legend.
Point 16:
- It is unclear which version of piggyBac was used in which mutagenesis protocol. Section 2.1. mentions “PBase gene”. Section 2.2. mentions “a mutant library of piggyBac transposase DNA”. Section 2.3. mentions “optimized-hyPBase as a template for mutation”.
Response 16: Thanks for your comments. We have marked the version of piggyBac clearly in manuscript as your suggestion. Briefly,the WT PBase was used to construct the screening system. The optimized hyPBase was used to mutagenize, screen, and get the hyperactive piggyBac transposase bz-hyPBase. The optimized WT PBase was used to prove that the screening system is suitable for other versions of piggyBac transposase.
Point 17:
- The plasmid maps of ploxP-PB, ploxP-PB-mutant, and pSAD-EGFP should be included in the manuscript, as are the plasmid maps of Fig. 1A and Fig. 1B. References for the plasmids used in the construction of all newly-constructed plasmids should be present in the manuscript.
Response 17: Thanks for your suggestion. The plasmid maps of pLoxP-optimized hyPBase and pSAD-EGFP are added in Figure S2. The references have added [33, 34, 35].
Point 18:
It is unclear what is in the upper and what in the lower row in Fig. 4B.
Response 18: Thanks. The upper and the lower row in Fig. 4B are two repeated result. We have added the description in legend.
Reviewer 2 Report
The manuscript covers an important area and presents methodology of screening hyperactive transposases that has a wide application range. The reviewer requests no further experimentation and recommends the manuscript for publication. but the manuscript just needed some editorial work of spelling, grammar etc.
Author Response
The manuscript covers an important area and presents methodology of screening hyperactive transposases that has a wide application range. The reviewer requests no further experimentation and recommends the manuscript for publication. but the manuscript just needed some editorial work of spelling, grammar etc.
Response: We thank you for your great comments. We have revised the manuscript as your suggestion and used a professional English editing service that recommended by MDPI. At the same time, we also add some supplementary materials to support the results in the manuscript. Thanks, again.
Round 2
Reviewer 1 Report
The authors have addressed my comments adequately. I have no further remarks.
Author Response
Thanks for your great comments. We appreciate the suggestions given by you, and we have benefited greatly from these suggestions. Thanks again.